# Clonal Interference and Mutation Bias in Small Bacterial Populations in Droplets

**DOI:** 10.3390/genes12020223

**Published:** 2021-02-04

**Authors:** Philip Ruelens, J. Arjan G. M. de Visser

**Affiliations:** Laboratory of Genetics, Wageningen University & Research, 6708 PB Wageningen, The Netherlands; philip.ruelens@wur.nl

**Keywords:** experimental evolution, antibiotic resistance, millifluidics, population size, mutation bias, selection bias

## Abstract

Experimental evolution studies have provided key insights into the fundamental mechanisms of evolution. One striking observation is that parallel and convergent evolution during laboratory evolution can be surprisingly common. However, these experiments are typically performed with well-mixed cultures and large effective population sizes, while pathogenic microbes typically experience strong bottlenecks during infection or drug treatment. Yet, our knowledge about adaptation in very small populations, where selection strength and mutation supplies are limited, is scant. In this study, wild-type and mutator strains of the bacterium *Escherichia coli* were evolved for about 100 generations towards increased resistance to the β-lactam antibiotic cefotaxime in millifluidic droplets of 0.5 µL and effective population size of approximately 27,000 cells. The small effective population size limited the adaptive potential of wild-type populations, where adaptation was limited to inactivating mutations, which caused the increased production of outer-membrane vesicles, leading to modest fitness increases. In contrast, mutator clones with an average of ~30-fold higher mutation rate adapted much faster by acquiring both inactivating mutations of an outer-membrane porin and particularly inactivating and gain-of-function mutations, causing the upregulation or activation of a common efflux pump, respectively. Our results demonstrate how in very small populations, clonal interference and mutation bias together affect the choice of adaptive trajectories by mediating the balance between high-rate and large-benefit mutations.

## 1. Introduction

In contrast to the common perception that evolution happens over long timescales, pathogens and cancer cells may change evolutionarily within a timescale of days. Such fast evolution can cause health problems when, for example, bacterial infections or cancer cells become resistant to the drugs used to cure them [1,2]. Predicting these unwanted evolutionary trajectories would present possibilities to intervene and control the problems they pose [3,4]. Although evolution is inherently associated with stochastic processes, such as the random occurrence and loss of mutations due to genetic drift and random changes in the environment, empirical evidence indicates that evolution can be surprisingly repeatable and, hence, may be predictable [5,6]. One fundamental and generic factor governing the repeatability of evolution is the size of the evolving populations [5,7,8]. As large populations suffer less from genetic drift and contain more adaptive variants than small populations, natural selection will likely filter out the small set of most beneficial mutants, even if they are rare, leading to more repeatable evolution through selection bias. By contrast, in small populations adaptation relies more heavily on the chance occurrence of beneficial mutations and, hence, mutation bias has a relatively stronger impact [9,10].

The repeatability in evolution has mostly been studied in laboratory evolution experiments with large well-mixed populations of asexual microbes [11,12,13,14], while in nature especially, populations of microbial pathogens often have much smaller effective populations sizes due to strong bottlenecks when colonizing new habitats or hosts or surviving drug treatment [15,16,17]. As such, it is surprising that very few studies of microbial adaptive laboratory evolution have been published with an effective population size smaller than 100,000 [3]. To determine the full diversity of possible evolutionary trajectories, as well as the interaction between mutation and selection bias in such small populations, it is, therefore, imperative to also investigate parallel evolution and its causes in very small populations. 

The standard method for performing evolution experiments involves serial passage of independently growing populations in a selective environment predefined by the researcher [13,14]. To permit continuous growth, cultures that have reached the size that the environment can sustain are diluted and transferred regularly to fresh medium. Traditionally, evolving populations have been maintained in culture tubes or microtiter plates; however, to maintain small effective microbial populations, it is necessary to use very small volumes or perform extreme back dilutions. Severe bottlenecks can, however, affect the mutational dynamics of evolving populations independent from the effect of population size by greatly reducing the genetic diversity and inhibiting fixation of beneficial mutations [18,19]. A promising new technology that might allow evolution in very small volumes is the use of millifluidic droplets as millimeter-scale bioreactors.

In the present project, we evolved isogenic strains of the bacterium *Escherichia coli* with a wildtype (WT) and a mutator phenotype in the presence of a constant low concentration of the β-lactam cefotaxime for ~100 generations at an effective population size of ~2.7 × 10^4^. To do this, a state-of-the-art droplet-based millifluidic device was utilized, which allowed semi-automated serial-passage evolution experiments of very small culture volumes (~0.5 μL) and population sizes [20]. By characterizing the evolved populations phenotypically and genotypically, we show that while adaptation of these small populations is limited by the supply of beneficial mutations, increasing the overall mutation rate by ~30-fold can alleviate this limitation and result in substantially increased adaptation through similar high-benefit mutations upregulating and activating a multidrug efflux pump, removing an outer-membrane porin, or increasing the production of outer-membrane vesicles. By contrast, evolution in populations with a wildtype mutation rate is restricted to the presumably most readily available beneficial mutations.

## 2. Materials and Methods

### 2.1. Strains and Medium

The *E. coli* B strains used in this study are REL606 and a mutator derivative of REL606, REL606 Δ*mutS*. REL606 Δ*mutS* has a deficient methyl-directed mismatch-repair function and was constructed by P1 transduction of a disrupted allele of *mutS* into REL606 [21], which resulted in ~34-fold higher mutation rate than REL606 [22]. Both strains were fluorescently labelled with yellow and blue fluorescent protein by inserting cat-J23101-SYFP2 or cat-J23101-mTagBFP2 into the *galK* gene using the Quick and Easy *E. coli* Gene Deletion Kit, but without removal of the chloramphenicol (cat) selection marker (Gene Bridges, Heidelberg, Germany). 

M9 medium used in this study contained 12.8 g/L Na_2_HPO_4_·H_2_O, 3 g/L KH_2_PO_4_, 0.5 g/L NaCl, 1 g/L NH_4_Cl, 2 mM MgSO_4_, 0.1 mM CaCl_2_, 0.5% (*w*/*v*) casamino acids (Gibco, Thermo Fisher Scientific, Waltham, MA, USA), 0.5 mM thiamine hydrochloride (Sigma-Aldrich, Saint Louis, MO, USA), and 8 g/L D-glucose. LB-agar was composed of 5 g/L yeast extract, 10 g/L trypticase peptone, 10 g/L NaCl, and 15 g/L agar.

### 2.2. Experimental Evolution

Overnight cultures inoculated with a single confirmed transformant of REL606 BFP, REL606 YFP, REL606 Δ*mutS* YFP, or REL606 Δ*mutS* BFP were plated on LB agar plates, from which 10 distinct colonies for each of the four strains were used as ancestors for the evolution experiment. These colonies were grown overnight in a 96-well microtiter plate with M9 medium supplemented with 20 mg/L chloramphenicol. The resulting culture was used to initiate the evolution experiment by diluting the culture 300-fold in fresh M9-medium supplemented with 0.032 mg/L cefotaxime in a 96-well plate. The selection cefotaxime concentration represented ¼ of the minimum inhibitory concentration (MIC) of WT and 1/6 of the Δ*mutS* strain as determined by the agar dilution method. This plate was placed inside the Millidrop Analyzer Azure system (Millidrop, France, http://www.millidrop.com), which generated 10 consecutive droplets of each population, separated by 10 droplets containing no cells, resulting in a train of millifluidic droplets of ~0.5 μL that was incubated for 16 h at 37 °C. After incubation, the droplets were removed from the machine and each set of 10 consecutive populations of the same genotype was collected in a separate well of a 96-well plate containing 50 μL of Millidrop carrier fluid. After 100 μL of non-selective M9 medium was added to the carrier fluid containing the droplets, the sorted droplets and M9 medium were mixed by pipetting up and down several times and transferred to a new 96-well plate. The populations were then allowed to grow overnight at 37 °C until saturation in non-selective conditions, after which they were 300-fold diluted in selective M9 medium that was then used to generate a new droplet train in the Millidrop system. In total, 12 transfers were performed. A single droplet can support ~10^5^ cells, and each population consisted of 10 consecutive droplets. Given a dilution factor of 300, we estimate that the effective population size was approximately 27,000 (≈ 333 cells (N_0_) × 8.22 generations between transfers × 10 pooled droplets). 

### 2.3. Whole Genome Sequencing and Analysis

Following the final passage in the Millidrop systems, population cultures were frozen as glycerol stock and streaked out on LB-agar plates supplemented with 20 mg/L chloramphenicol. A single colony was randomly selected and used to inoculate 2 mL of growth medium. Genomic DNA was extracted from the overnight culture in non-selective M9 medium using a Gentra Puregene Yeast/Bact. DNA extraction kit according to the manufacturer’s instructions (Qiagen, Hilden, Germany). DNA integrity and concentration were assessed by gel electrophoresis and a Qubit3.0 fluorometer, respectively (Thermo Fisher Scientific Inc., Waltham, MA, USA). Library preparation and whole-genome sequencing of the evolved clones was performed by Novogene using an Illumina Novaseq 6000 with the PE150 platform and an average coverage of ~300 fold. Reads were analyzed using the Breseq computational pipeline to predict single-nucleotide polymorphisms, indels, and structural variants [23]. The raw sequencing data are available upon request.

### 2.4. Inferring Independent Mutational Events

Using the genomic data, we inferred a phylogenetic tree of the evolved clones using the phylogenetic and analysis package “phangorn” in R [24]. First, a distance matrix was calculated using the sequencing data. From the distance matrix, a neighbour-joining tree was generated, which was used as a start tree to search for the most parsimonious tree using the parsimony ratchet method. Subsequently, the accelerated transformation algorithm was used to infer parsimony branch lengths upon the most parsimonious tree. Finally, the reconstructed phylogeny was manually curated to not allow reversal of mutation, which we deemed very unlikely considering adaption was only allowed for 100 generations.

### 2.5. Minimum Inhibitory Concentration

The MIC of cefotaxime (CTX) of the evolved populations and selected clones was determined using a non-standard agar dilution method. First, M9 agar plates supplemented with increasing concentrations of cefotaxime were prepared (0, 0.032, 0.048, 0.064, 0.096, 0.128, 0.192, 0.256, 0.384, and 0.512 mg/L [CTX]). From overnight culture in non-selective M9 medium, ~2 × 10^5^ cells (~10 × N_e_) were spotted on the surface of the agar plates and subsequently incubated for 24 h at 37 °C. The MIC was defined as the highest cefotaxime concentration where no confluent lawn of growth was visible.

### 2.6. Growth Rate Measurements

Sequenced end clones and ancestral clones were inoculated from glycerol stocks into M9 minimal media supplemented with 20 mg/L chloramphenicol and grown overnight at 37 °C with shaking. In a 96-well plate, 200 µL M9 medium without cefotaxime and selective M9 medium per well was then inoculated in triplicate at a 1:100 dilution from the overnight culture. To reduce evaporation and condensation, the samples were covered with 50 µL mineral oil. The 96-well plate was placed in a Victor3 microtiter plate reader (PerkinElmer, Waltham, MA, USA), and absorbance was measured every three minutes for 16 h at 37 °C with 30 s of vigorous shaking every three minutes before measuring. Growth rates represent the maximum linear slope of the natural logarithm of absorbance versus time in a 2 h window. 

### 2.7. Statistical Analysis

All statistical analyses were performed using R version 3.6.1 [25]. For comparing MIC and growth rates, the non-parametric Mann–Whitney U test was used as implemented in R. General linear model diagnostics were conducted using the “DHARMa” package in R [26].

## 3. Results and Discussion

To allow evolution of very small populations in millimetric droplets, we utilized the droplet-based Millidrop Analyzer Azure system [20], where every droplet represents an independent closed-off culture of approximately 0.5 μL (Figure 1A,B). This system permitted us to collect the contents of individual droplets following incubation, which allowed us to use it as a serial-transfer “evolution machine”.

Experimental evolution was performed in the presence of a sub-lethal concentration of β-lactam antibiotic cefotaxime with *Escherichia coli* B strain REL606, which has been regularly used for experimental evolution [11,12,22]. In order to uncouple the effects of population size on mutation supply from those on selection and drift, a mutator version of the same strain was also used. This variant carries a disrupted copy of the DNA mismatch-repair gene *mutS*, resulting in an approximate 30-fold increase of its mutation rate [22]. Both WT and mutator strains (MutS) were labeled with a yellow or blue fluorescent protein in order to track growth and possible interpopulation contamination throughout the evolution experiment.

In total, 20 WT and 20 MutS populations underwent 12 transfers, which corresponded to ~100 generations. To evaluate the extent of short-term adaptation within the droplet populations, the minimum inhibitory concentration (MIC) of cefotaxime of ancestors and evolved populations was determined using an agar dilution method. These results showed that the populations that originated from WT and MutS could, on average, grow on cefotaxime concentrations 2.7- and 2.4-fold higher than their ancestral population, respectively (Figure 1C, Appendix A). Unfortunately, when assessing cross-contamination of the end populations using fluorescent flow-cytometry, we observed pervasive contamination throughout the experiment, which likely originated during the sorting of the droplets (personal communication with Millidrop company).

To reveal the genetic basis for increased cefotaxime resistance in the evolved droplet populations, a single clone of each population was randomly selected and subjected to whole-genome resequencing. Single nucleotide polymorphisms (SNPs), small indels, structural variants, and IS1-mediated mutations were identified using the breseq computational pipeline (see methods, [23]). Both WT and MutS cells invaded other populations, as 10 MutS clones were isolated from WT started populations and 11 WT clones from initially MutS populations. Since contamination hampered the unambiguous detection of parallel evolution, we inferred the phylogenetic relationships of the forty evolved clones to distinguish between shared mutations that originated from independent mutational events and shared mutations due to common ancestry and migration between populations (Figure 2). Overall, we identified 13 mutations in end clones from the WT populations and 433 mutations in clones from the MutS populations (on average 0.61 mutations per WT versus 22.8 mutations per MutS clone, thus 37-fold higher in MutS clones, comparable with their ~30-fold higher mutation rate). Notably, eight of the sequenced WT clones did not contain any mutations, indicating that although the population MIC significantly increased for all evolved populations, fixation of mutations did not occur in all populations. All identified mutations per clone are shown in Appendix A.

Since sequencing allowed us to unambiguously designate which clone descended from a WT or MutS ancestor, we determined the extent of cefotaxime adaptation of all evolved clones by assessing their growth rate in selective medium. The growth rate of MutS clones increased on average more than three-fold relative to their ancestor, while WT clones did not significantly increase their growth rate (Figure 3A). To determine whether the increased growth rate of MutS clones in the presence of cefotaxime came at a fitness cost in the absence of antibiotic—either due to linked deleterious mutations or to costly resistant mutations—we also assessed the growth rate in the absence of cefotaxime. While the WT clones increased their growth rate in the absence of antibiotic, MutS clones on average did not but showed great variation in their growth rate across clones (Kruskal–Wallis test: growth rate at [CTX] = 0 ~ MutS clone: *R*^2^ = 0.010, *p* = 0.66 χ^2^ = 51.62, df = 18, *p* < 0.0001, Figure 3B, Appendix A). Interestingly, the growth rate of MutS clones, both in the absence and in the presence of cefotaxime, exhibited a significant linear dependence on the number of mutations present in these clones (Figure 3C), while for WT clones, no such relationship was apparent (growth rate at [CTX] = 0 ~ number of mutations: *R*^2^ = 0.010, *p* = 0.66; growth rate at [CTX] = 0.032 μg/mL ~ number of mutations: *R*^2^ = 0.017, *p* = 0.57). This shows that adaptation was limited by the supply rate of mutations, not only in the WT populations but even in the MutS populations with approximately 30-fold higher mutation rates, as might be expected for such small populations [22].

We identified parallel mutations in four genes related to three distinct antibiotic resistance targets (Figure 2, inset). Two of the identified genes (*acrR* and *acrB*) were associated with the AcrAB-TolC multidrug efflux pump. *AcrB* encodes a subunit of the pump that acts as an antiporter depending on drug affinity [27], while *acrR* negatively regulates the expression of the *acrAB* operon [28]. In MutS clones, eight non-synonymous SNPs were identified in *acrB*, and in *acrR*, two non-synonymous SNPs and one frame-shift indel were detected. Considering that AcrB is a pivotal subunit of the efflux pump, the non-synonymous mutations therein likely constitute gain-of-function mutations that increase the affinity of AcrB for cefotaxime. Another frequently mutated gene is *nlpI*, which encodes an outer-membrane anchored lipoprotein and is linked to the production of outer-membrane vesicles (OMV) [29]. Loss-of-function mutants of *nlpI* exhibited increased OMV production, resulting in enhanced resistance against antibiotics [30,31]. Out of the 11 mutations in *nlpI*, 10 were either single-nucleotide indels or introduced a premature stop codon and hence were predicted to cause an increased production of OMVs. The final gene that mutated multiple times was *ompR*, a pleiotropic regulator controlling the expression of multiple genes, including the positive regulation of outer-membrane porin OmpF involved in the passive transport of antibiotics [32,33]. *OmpR* was mutated three times in three different MutS clones by non-synonymous SNPs, which presumably lowered OmpF expression. 

Of the aforementioned resistance targets, only two independent loss-of-function mutations in *nlpI* were identified in WT clones. Moreover, both *nlpI* mutations successfully spread to another droplet population (see Figure 2), consistent with their adaptive benefit. MutS clones exhibited a greater variation in resistance targets affected by mutation. Besides mutations inactivating *nlpI*, as in WT clones, MutS clones harbored additional loss and gain-of-function mutations associated with the upregulation and activation of efflux and downregulation of influx. In a recent experimental evolution study with the same *E. coli* strain using increasing concentrations of cefotaxime and 100- and 10,000-fold larger populations than used in this study, the same resistance targets were identified, among others [10]. 

To assess the relative selective benefit of the mutations in common resistance targets, we estimated their effects on the growth rate in the presence and absence of cefotaxime using a general linear model under the assumption of additive mutation effects (Figure 3D). This analysis showed that in the presence of antibiotics, mutations in *acrR*, *acrB,* and *nlpI* and mutations in other genes all significantly increased the growth rate, while no significantly positive effect could be attributed to *ompR* mutations. Moreover, mutations affecting *acrR* and *acrB* had a roughly 40% larger benefit (albeit not significantly larger, Wald test: *p* = 0.52) than mutations in *nlpI*. In the absence of antibiotics, only *nlpI* mutations significantly decreased the growth rate, suggesting that the benefit of *nlpI* mutations is limited by pleiotropic fitness costs.

Considering that the only resistance target affected in more than one WT clone was *nlpI*, we hypothesize that these mutations occur at a higher rate than mutations affecting *acrR*, *acrB*, or *ompR*, which arise in MutS populations, where they either drive down smaller-benefit *nlpI* mutations that are also present or hitchhike on their background. Furthermore, a mutation that is present in parallel in three WT clones (clones 5, 7, and 21; see Figure 2), which does not affect a known resistance target, is a large structural IS1-mediated deletion starting at *cybB*. This IS1-mediated mutation has been previously identified in evolution experiments and mutation accumulation studies using the same *E. coli* strain but other conditions, consistent with its high rate and lack of involvement in cefotaxime resistance [11,34]. The fact that the same deletion is observed in only one MutS clone further suggests that competition with larger-benefit mutations in the MutS populations limited their contribution and shaped the adaptive trajectories.

## 4. Conclusions

In this study, we have shown that short-term adaptation to a novel antibiotic of droplet populations with an effective size of approximately 27,000 bacteria is limited by the supply rate of beneficial mutations, both for cells with wildtype mutation rate and cells with a 30-fold higher mutation rate. Yet, we observed parallel evolution at the gene level even in these small wildtype populations due to two types of mutations with a presumably high-rate: one associated with increased resistance due to the loss-of-function of a lipoprotein causing higher production of outer-membrane vesicles, the other an IS1-mediated deletion with likely no effect on cefotaxime resistance. In populations lacking proper DNA-mismatch repair, the increased mutation supply rate allowed for adaptation through rarer high-benefit mutations, including gain-of-function SNPs enhancing the affinity of an efflux pump for cefotaxime. Our results demonstrate short-term adaptive benefits of mutator mutants in very small populations, driven by the selection of more, as well as larger-benefit, mutations. Moreover, our findings show that clonal interference among diverse, available, beneficial mutations can mediate the relative contribution of high-rate and large-benefit mutations, even in very small bacterial populations, with potential implications for long-term adaptation. These findings support and extend results from a recent study with the same strain expressing a plasmid-borne β-lactamase, demonstrating different mutation choices in populations about 100- to 10,000-fold larger in size than the droplet populations [10], as well as a study of quinolone resistance in *E. coli* K12 in large populations with varying bottleneck size [17].

## Figures and Tables

**Figure 1 genes-12-00223-f001:**
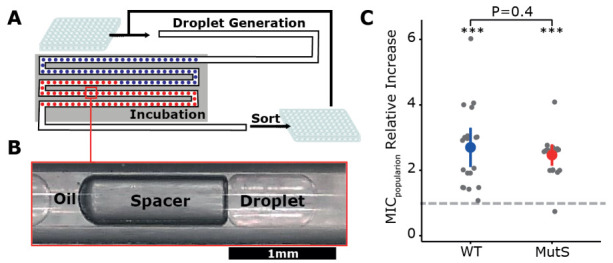
Using millifluidics for experimental evolution. (**A**) Schematic representation of the millifluidic experimental evolution design. (**B**) Within the device up to 960 millimetric droplets of ~0.5 µL each form a train separated by an air-filled spacer bubble and oil. (**C**) Population minimum inhibitory concentration (MIC) fold increase relative to their respective ancestor (dashed line) after 100 generations in selective medium. For both wildtype (WT) and mutator strain (MutS) populations, the relative increase of MIC was significantly greater than one at *p* < 0.001 (***, Mann–Whitney U test). The average increase in MIC did not significantly differ between WT and MutS populations (*p* = 0.4, Mann–Whitney U test).

**Figure 2 genes-12-00223-f002:**
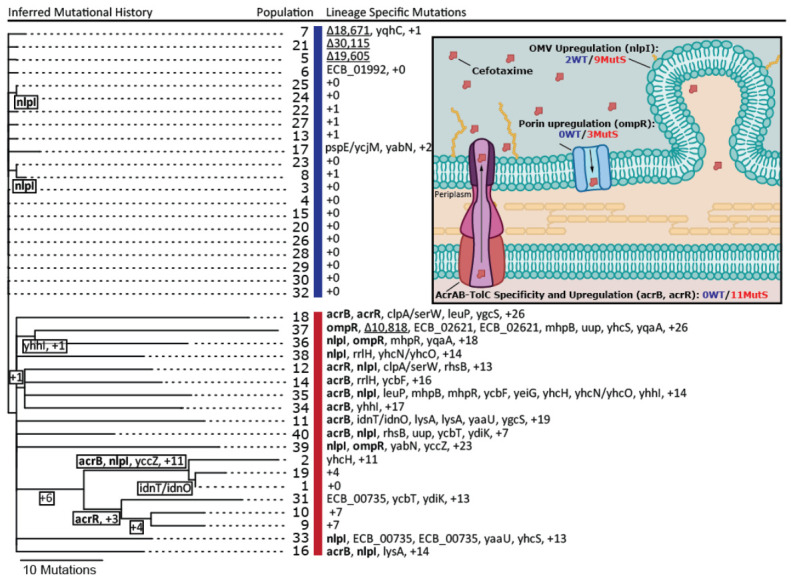
Inferred phylogenetic tree of the 40 sequenced end clones. Only parallel mutations that occurred in at least two clones independently are shown by their gene name. Mutations that occurred at least two times independently in known resistance genes are highlighted in bold. The recurring IS1-mediated deletion, starting at cybB, is underlined. Intergenic mutations are indicated by a slash in between the two genes surrounding this mutation. WT clones are indicated by the blue vertical bar, MutS clones by the red bar. The inset shows the cefotaxime resistance mechanisms identified in the evolved clones and the total number of mutations in each resistance mechanisms in WT and MutS clones. The named genes in brackets show the main genes mutated for each mechanism. OMV: outer-membrane vesicles.

**Figure 3 genes-12-00223-f003:**
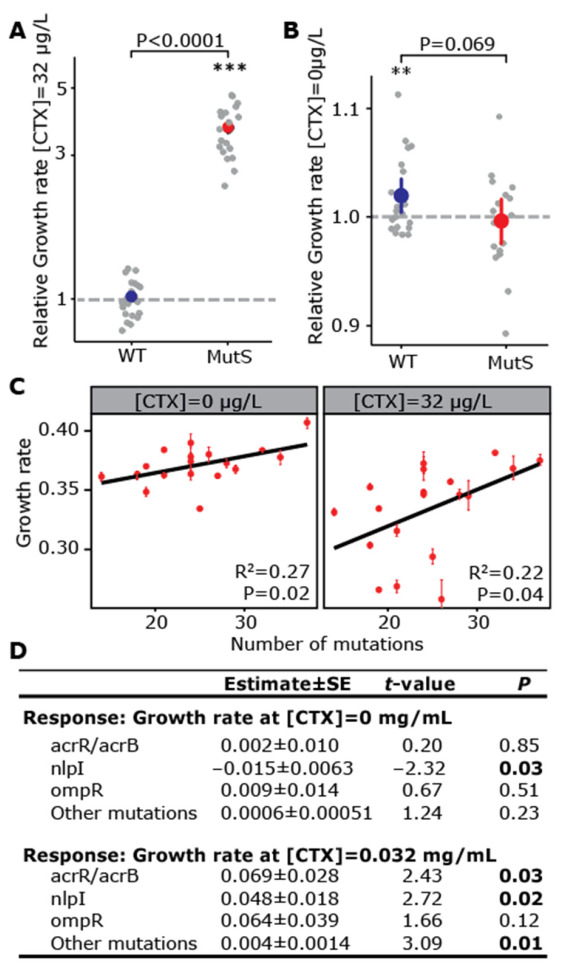
Fitness measurements of clones and fitness effects inferred for specific mutations in the presence and absence of cefotaxime. Growth rate in the absence (**A**) and presence of cefotaxime (**B**) of the sequenced clones relative to their respective ancestor (dashed line). Asterisks indicate significant increase (Mann–Whitney U test, **: *p* < 0.01, ***: *p* < 0.001). The *p*-values shown on top are from a Mann–Whitney U test comparing WT and MutS clones. (**C**) Linear regression analysis between the number of mutations in sequenced MutS clones and their relative growth rate in the absence and presence of cefotaxime. Error bars represent the standard error of the mean of three biological replicates. (**D**) Selection coefficient estimates and significance results of common resistance mutations, inferred using a general linear model of the growth rate in the absence and presence of cefotaxime as a linear function of the number of mutations in each category. *p*-values in bold are significant at *p* < 0.05 level.

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
