# Peer review of "Clonal Interference and Mutation Bias in Small Bacterial Populations in Droplets"

_genes, 2021, doi:10.3390/genes12020223_

Round 1

Reviewer 1 Report

The manuscript titled “Clonal interference and mutation bias in small bacterial populations in droplets ” describes the selection of E. coli populations in millidroplets where the effective population size during serial transfers is kept very low. The paper has a sound underlying reasoning, is in general well conducted and reported. However  I do have some issues understanding some of the presented data which needs to be addressed. Find some questions/more detailed comments below:

Title: Given the title I find that the aspect of clonal interference and mutation bias should come back more explicitly in the discussion.

Line 20: odd sentence

Fig 3: how do the authors explain an increase in growth rate in conditions without antibiotic (fig 3c)? Can they speculate how this would work? Does antibiotic resistance always come with higher growth rates when exposed to sublethal concentrations or could one imagine that slower growth could have an advantage (could one enrich for such organisms in the compartments they used)? Does the addition of sublethal antibiotic concentrations affect the final OD?

Fig 3: please show standard errors for growth rates of the individual strains. This seems especially relevant in panel c where the authors claim a linear relationship between the number of mutations and higher growth rates (I see that the p value is significant but wonder about the biological meaning of such a low r2). To me it is seems discrepant that in 3b there is no growth rate effect for the mutS strain whereas in 3c the authors claim it increases in time (line 219 onwards). I also don’t understand how the relative growth rates in 3a can increase 3x while in 3c it goes from ~0.3 to <0.40

Line 198-200: could genome rearrangements play a role here? Was that assessed in the sequencing data (might be difficult if reads are short)?

Given the small population size and the limited increase in growth rate of individual mutations in the evolved isolates I am surprised that so many mutations accumulated after only 100 generations. Does this result match with  the mutation rates of these strains. My back of the envelope calcuation suggests that very few mutations should occur in such  small populations which makes the likelihood of picking up beneficial mutations very low - but maybe I am using incorrect mutation rates. Can the authors elaborate a bit more on this and put things into perspective?

Author Response

Point-by-point response to the reviewer’s comments

We would like to thank the reviewers and editors for their time and effort in assessing our manuscript. Below, we provide a detailed point-by-point response to the comments and suggestions of both reviewers.

Reviewer 1 

Title: Given the title I find that the aspect of clonal interference and mutation bias should come back more explicitly in the discussion.

We thank the reviewer for this comment, and have now changed the final sentence of the abstract, as well as text in the conclusion section to more explicitly refer to clonal interference and mutation bias as factors determining the mutation choices also in these very small bacterial populations.

Line 20: odd sentence

 We agree with the reviewer that this sentence was confusing and the sentence has been rewritten in the revised manuscript.

Fig 3: how do the authors explain an increase in growth rate in conditions without antibiotic (fig 3c)? Can they speculate how this would work?

By evolving in sublethal antibiotic concentrations, the populations might not only be evolving increased resistance, but concomitantly increased growth rate in the minimal medium used during experimental evolution, which could explain the increased growth rates in conditions without antibiotics.

Does antibiotic resistance always come with higher growth rates when exposed to sublethal concentrations or could one imagine that slower growth could have an advantage (could one enrich for such organisms in the compartments they used)?

Decreased growth rates can indeed have a survival advantage in the presence of beta-lactam antibiotics as an increased growth rate typically also increases the lysis rate by beta-lactam antibiotics [1]. Bacterial persistence represents an extreme example of this strategy. However, during evolution in sublethal concentrations, such a strategy is likely not viable as they would be outcompeted by faster-growing WT cells. More importantly, evolving at sublethal concentrations possibly prevents large effect but very costly resistance conferring mutations to be selected.

Does the addition of sublethal antibiotic concentrations affect the final OD?

Sublethal antibiotics decreased the final OD by on average 15% across all evolved lines.

Fig 3: please show standard errors for growth rates of the individual strains. This seems especially relevant in panel c where the authors claim a linear relationship between the number of mutations and higher growth rates (I see that the p value is significant but wonder about the biological meaning of such a low r2).

The standard errors for the growth rates of individual strains in the absence and presence of cefotaxime can already be found in Supplemental Table 1. For the reader’s convenience and as per reviewer’s suggestion, we have also added them to Fig 3c.

To me it is seems discrepant that in 3b there is no growth rate effect for the mutS strain whereas in 3c the authors claim it increases in time (line 219 onwards).

Fig 3B shows that the relative growth rate of WT and MutS sequenced lines in the absence of antibiotic are not significantly different. However, 3C represents the relationship of relative growth rate and the number of mutations for only MutS clones at both selective and non-selective conditions. Moreover, we do not claim that the growth rate increases in time, but our data does show that their growth rate is linearly dependent on the number of mutations, suggesting adaptation was mutation-limited. Although time and the number of mutations are likely correlated, this does not mean that time or rather the length of the experiment is correlated with the growth rate.

 I also don’t understand how the relative growth rates in 3a can increase 3x while in 3c it goes from ~0.3 to <0.40

The relative growth of MutS (Fig 3a) indeed increased approximately 3-fold to 0.3-0.4 as in 3c which is due to the low growth rate of the MutS ancestor in the presence of cefotaxime (+- 0.09).

Line 198-200: could genome rearrangements play a role here? Was that assessed in the sequencing data (might be difficult if reads are short)?

Genome rearrangements could indeed play a role here and given the ability of the breseq computational pipeline to identify large-scale chromosomal changes from even short-read data [2], we would expect such rearrangements to be identified, especially given the high coverage of our samples (~300x). The most likely explanation here is that the beneficial mutant or mutants within the population did not achieve high frequencies at the end of the evolution experiment and by randomly selecting a clone for sequencing we missed these resistance conferring mutations.

Given the small population size and the limited increase in growth rate of individual mutations in the evolved isolates I am surprised that so many mutations accumulated after only 100 generations. Does this result match with  the mutation rates of these strains. My back of the envelope calculation suggests that very few mutations should occur in such  small populations which makes the likelihood of picking up beneficial mutations very low - but maybe I am using incorrect mutation rates. Can the authors elaborate a bit more on this and put things into perspective?

In Perfeito et al. [3], they estimate the beneficial mutation rate for populations with an effective size similar to the population size used in our study to be 2 x 10-5 per genome, per generation. Such a beneficial mutation rate would mean that on average every other generation an adaptive mutant arises in the evolving small WT populations. In the MutS population with an estimated roughly 30-fold increased mutation rate, this translates to approximately 16 beneficial mutations in the population every generation. Of course not all mutations will establish and reach high allele frequencies due to initial drift extinction, but this estimation seems congruent with on average 0.61 mutations per sequenced WT clone and 22.8 mutations per MutS sequenced clone at the end of the evolution experiment (i.e. ~37-fold difference).

References

  1. Lee, A.J., et al., Robust, linear correlations between growth rates and β-lactam–mediated lysis rates. Proceedings of the National Academy of Sciences, 2018. 115(16): p. 4069-4074.
  2. Barrick, J.E., et al., Identifying structural variation in haploid microbial genomes from short-read resequencing data using breseq. BMC Genomics, 2014. 15(1): p. 1039.
  3. Perfeito, L., et al., Adaptive Mutations in Bacteria: High Rate and Small Effects. Science, 2007. 317(5839): p. 813-815.

Reviewer 2 Report

In this study, the authors use droplets to grow only few E. coli cells in evolutionary experiments, using an antibiotic in sub-inhibitory concentrations. Indeed, the mutation supply seems to be most critical for (very) small population sizes.

I think it's a nice study. It looks like they've done everything I would want.

Minor: 

Maybe the authors could expand a little more on connected medical issues. I believe this would attract more readers (just to mention one example which came to my mind:  PMID 16048951).

µL is not printing in the abstract.

Chloramphenicol is sometimes capitalized, sometimes not. 

Raw sequencing data should be submitted.

Author Response

Point-by-point response to the reviewer’s comments

We would like to thank the reviewers and editors for their time and effort in assessing our manuscript. Below, we provide a detailed point-by-point response to the comments and suggestions of both reviewers.

Reviewer 2

Maybe the authors could expand a little more on connected medical issues. I believe this would attract more readers (just to mention one example which came to my mind:  PMID 16048951).

While we appreciate the reviewer’s suggestion, further discussion concerning medical relevance, beyond what is mentioned in the introduction (i.e. the importance of population bottlenecks for pathogens), is outside of the scope of this study or too speculative in our opinion.

µL is not printing in the abstract.

Thanks for pointing this out. This has been corrected.

Chloramphenicol is sometimes capitalized, sometimes not. 

Corrected.

Raw sequencing data should be submitted.

The data will be made available. We’ve added a sentence in the methods that it is available upon request but we will discuss with the editor whether they prefer a specific online data repository.